# Use of Melatonin and/on Ramelteon for the Treatment of Insomnia in Older Adults: A Systematic Review and Meta-Analysis

**DOI:** 10.3390/jcm11175138

**Published:** 2022-08-31

**Authors:** Srujitha Marupuru, Daniel Arku, Ashley M. Campbell, Marion K. Slack, Jeannie K. Lee

**Affiliations:** R. Ken Coit College of Pharmacy, The University of Arizona, Tucson, AZ 85721, USA

**Keywords:** melatonin, insomnia, sleep outcomes, and older adults

## Abstract

To investigate the efficacy of melatonin and/or ramelteon reporting sleep outcomes for older adults with chronic insomnia, a systematic review and a meta-analysis of PubMed, EMBASE, Cochrane library, International Pharmaceutical Abstracts, PsycINFO, science citation index, center for reviews and dissemination, CINAHL, grey literature and relevant sleep journal searches were conducted from 1 January 1990 to 20 June 2021. Randomized controlled trials and other comparative studies with melatonin and/or ramelteon use among older patients with chronic insomnia were included. Funnel plot and Egger’s test was used to determine publication bias. A forest plot was constructed to obtain a pooled standardized mean difference using either a fixed or random effects model for each of the two broad categories of sleep outcomes: objective and subjective. Of 5247 studies identified, 17 studies met the inclusion criteria for MA. Study sample size ranged from 10 to 829 with the mean age ≥55 years. There were significant improvements in total sleep time (objective), sleep latency and sleep quality (objective and subjective) for melatonin and/or ramelteon users compared with placebo. Sleep efficiency was not significantly different. The effects of these agents are modest but with limited safe treatment options for insomnia in older adults, these could be the drugs of choice.

## 1. Introduction

Sleep disorders affect approximately 20% of the American population [1]. According to the American Academy of Sleep Medicine, insomnia is a common sleep disorder that is defined as “the subjective perception of difficulty with sleep initiation, duration, consolidation, or quality that occurs despite adequate opportunity for sleep, and that results in some form of daytime impairment” [2]. Categorized further, chronic insomnia is a condition of disrupted sleep that occurs at least three nights per week and lasts for at least three months [3]. One third of adults will subjectively describe problems falling asleep, remaining asleep, or awakening too early [4]. Insomnia adversely affects older adults at a rate higher than that in the general adult population [5]. There are several important physiological processes that occur to create age-related changes in sleep that spans over a lifetime. As a person ages, the average amount of sleep time decreases from being 6.5 to 8.5 h per night in a young adult to 5 to 7 h of sleep per night in older adults [6]. This finding highlights the fact that the maximal sleep capacity, as indicated by total sleep time (TST), decreases with age. Younger people are reported to have a maximal sleep capacity of 8.9 h versus 7.4 h in older people [7]. The older population aged 65 years and older will be the largest segment affected by insomnia in the next 20 years [8]. Thus, the appropriate diagnosis and treatment of insomnia are paramount among older adults, especially because they are often more vulnerable to the adverse effects of common treatments [5].

Benzodiazepines and nonbenzodiazepine benzodiazepine-receptor agonists have a long history of use in insomnia due to their hypnotic properties, yet they lead to minimal improvements in sleep latency and duration [9]. Furthermore, their uses have several drawbacks including an increased risk of cognitive impairment, delirium, falls and fractures, motor vehicle accidents; residual (‘hangover’) effects, particularly with longer-acting agents; and the potential to cause withdrawal effects, dependence, and tolerance [9]. Nonbenzodiazepine benzodiazepine-receptor agonists have also been associated with increased emergency department visits and hospital admissions [9]. Research aimed at developing safer and more efficacious hypnotic agents for use in older adults has received urgent attention. Therefore, previous studies have highlighted the potential use of melatonin in treating primary [10] and secondary [11] sleep disorders in adults. However, studies reveal inconsistent findings, with some of them reporting beneficial effects of melatonin on sleep, while others document only marginal effects.

Melatonin is a hormone produced by the pineal gland from the essential amino acid tryptophan (N-acetyl-5 methoxytryptamine) [12]. There are two forms of oral melatonin; simple melatonin with short half-life and prolonged release melatonin with longer half- ife but neither have been found to be superior for the treatment of insomnia [13]. The use of exogenous melatonin has not been reported to be associated with tolerance, dependence, or a ‘hangover effect.’ In addition, it has minimal side effects (e.g., headache, dizziness, nausea) if administered at a low dose [14], which could significantly affect quality of life in older adults. Varying doses and formulations of melatonin are available over the counter without a prescription. A melatonin receptor agonist, ramelteon, was approved in the United States in 2005 as a prescription treatment for insomnia [15]. Ramelteon is a novel melatonin MT1 and MT2 receptor subtypes agonist that has specific effects on melatonin receptors in the suprachiasmatic nucleus [16]. Clinical investigations and post-marketing surveillance of ramelteon have found this drug be to be safe in patients with insomnia [17].

A meta-analysis conducted by Kuriyama et al. [18] on 13 studies involving 5812 subjects concluded that ramelteon was associated with improvement in TST and sleep efficiency (SE). However, it is important to note that this analysis included randomized controlled trials (RCTs) involving ramelteon only, with the last search date in 2012 and mean participant age of 48 years. Another meta-analysis, with the search ending in 2012, included RCTs only to evaluate the use of melatonin for treating primary sleep disorders in children and adults [19]. This study demonstrated that melatonin decreased sleep latency (SL), increased TST and improved overall sleep quality. Nevertheless, there is a current knowledge gap concerning the effects of melatonin use on sleep outcomes in an older population. Hence, a comprehensive systematic review and meta-analysis may strengthen the evidence of melatonin and/or ramelteon as a treatment option for older adults with insomnia. Because the diagnosis of insomnia generally originates with a subjective complaint of poor sleep, assessing subjective parameters along with objective measures is necessary for optimal evaluation.

Based on the existing evidence and identified gaps, the aim of the current systematic review and meta-analysis was to assess the latest evidence of melatonin and melatonin receptor agonist (ramelteon) use in older adults ≥50 years for the management of chronic insomnia. Specifically, we evaluated the effects on TST, SL, SE, and subjective sleep quality.

## 2. Materials and Methods

### 2.1. Study Selection

Inclusion and exclusion criteria for the studies followed the PICOS (Population, Intervention, Comparison, Outcomes and study design) framework. To be included in the systematic review, studies needed to use an RCT, prospective or retrospective cohort, case-control, or other observational study design with comparative groups that tested the effects of melatonin and/or ramelteon on insomnia outcomes in older adults. Though the geriatric population is often defined as those aged 65 years and older, we included studies reporting a mean age of 50 years or older to broaden search results and include applicable patient segments. Studies reporting quantitative outcomes by objective and/or subjective measures of insomnia were included. Along with duplicate references, non-English language publications, narrative reviews, systematic reviews, meta-analyses, and non-research publications were excluded. Studies that focused exclusively on other drug exposures or non-pharmacological interventions used for insomnia and pharmacokinetic studies were also excluded.

To be included in the present meta-analysis, studies had to assess effects of melatonin and/or ramelteon on insomnia outcomes. Insomnia-related outcomes included in the meta-analysis were TST, SL, SE, and subjective sleep quality. The studies had to report adequate data to calculate a standardized mean difference (SMD) to be included in the meta-analysis. This systematic review followed the Preferred Reporting Items for Systematic Reviews and Meta-Analyses (PRISMA) criteria.

### 2.2. Search Strategy

A comprehensive review of the literature, supplemented by additional grey literature and relevant sleep journal article searches, was conducted by two independent reviewers under the guidance of a medical librarian with expertise in systematic review searches. A search, from 1 January 1990 to 20 June 2021, was performed using NLM PubMed/MEDLINE, Ovid/MEDLINE, EMBASE, International Pharmaceutical Abstracts, PsycINFO, Cochrane Database of Systematic Reviews, SCOPUS, and CINAHL. In addition to these electronic database searches, grey literature such as clinicaltrials.gov, sleep associated websites, google scholar, and hand searching of review articles and meta-analyses was conducted. The year 1990 was chosen as the start of the search timeframe because the original Beers Criteria was published in 1991, which classified many of the classically used sleep medications as potentially inappropriate in older adults [9]. The keyword and terms used in the search were found anywhere in the citation and included insomnia, melatonin, ramelteon, sleep quality, and sleep time. Synonyms of these keywords, in addition to controlled vocabulary for MEDLINE, as well as terms related to measurement outcomes of insomnia such as “polysomnography”, “actigraphy”, “Electrooculography”, “Electroencephalography”, “Leeds sleep evaluation questionnaire”, “Pittsburgh Sleep Quality Index”, “Insomnia Severity Index”, “sleep diary”, were included in the search strategy.

### 2.3. Data Extraction

A dual review process was used for study inclusion and data extraction, where a team of two investigators reviewed study titles, abstracts, and full-text articles independently and met to combine the data by consensus. A third reviewer with clinical expertise in geriatric practice served as an arbiter when consensus was unable to be reached. For data extraction, a previously tested standardized form was adapted to this study and used to minimize variability. Data extraction consisted of study characteristics, patient characteristics, intervention and comparator information, method of measurement and outcome measures.

### 2.4. Outcomes

We used the SMD as the primary outcome, which was calculated for each included study. Conceptually, the SMD is a measure of the difference in effect between melatonin and/or ramelteon and the placebo. The size of the SMD can be interpreted as being small (<0.2), moderate (0.2–0.8), or large (>0.8) [20] The objective sleep outcomes: TST, SL and SE measured by polysomnography [PSG], actigraphy, electrooculography, and/or electroencephalography [EEG]. The subjective sleep outcomes were TST and SL measured by a sleep diary, sleep index, sleep scales/sleep log or sleep questionnaires: Pittsburgh Sleep Quality Index (PSQI), Leeds Sleep Evaluation Questionnaire (LSEQ), and/or Quality of sleep questionnaires (QOS).

### 2.5. Data Analysis

Descriptive and demographic data were summarized by calculating the overall means and standard deviations for continuous data, and categorical data are presented with frequencies and percentages. The results were synthesized by constructing a forest plot using a fixed or random effects model as appropriate for each of the two broad categories of outcomes: objective and subjective sleep outcomes. Heterogeneity was assessed using I^2^ statistics, which report the percentage of variability due to factors other than random variation. I^2^ values of greater than 50% are substantial and a random effect meta-analysis is run to control for this statistical heterogeneity [21] or otherwise a fixed effect meta-analysis is conducted. Funnel plots and Kendall’s tau were used to assess publication bias. Data analysis was performed using Comprehensive Meta-analysis Program software (Biostat, Englewood, NJ, USA).

Since most of the studies included in the systematic review used an RCT design, potential for bias was assessed using the RoB 2: A revised Cochrane risk-of-bias tool for randomized trials tool [22]. The following domains of the Cochrane Collaboration’s tool were selected to evaluate the risk of bias: arising from the randomization process; from assignment to intervention; from missing outcome data; measurement of the outcome; and selection of the reported results. The overall risk of bias for each trial was reported as ‘low’, ‘some concerns’ or ‘high’ based on these previous domains.

## 3. Results

### 3.1. Characteristics of the Included Studies

The database searches identified 9247 records. After removing duplicates, 5247 titles/abstracts were screened, and 69 full-text articles were subsequently assessed for eligibility. Of these full-text articles, 48 were excluded due to the exclusion criteria, and 21 studies [13,23,24,25,26,27,28,29,30,31,32,33,34,35,36,37,38,39,40,41,42]) were included in the final systematic review and 17 in the meta-analysis (Figure 1). 

Table 1 shows characteristics of the studies included in the systematic review. There were 16 RCTs, among which three used crossover study design, and five an open-label study. A total of 2462 subjects were involved in the 17 studies included in meta-analysis. Sample size ranged from 20 to 829 with the mean age of all included studies being 55 years and older, thirteen studies involved majority female patients, and eight studies reported comorbid conditions. Most of the studies (57%) were conducted in outpatient settings (*n* = 12), followed by long term care settings (*n* = 2). Melatonin doses ranged from 0.3 mg to 6 mg in fourteen studies, while ramelteon doses ranged from 4 mg to 8 mg in seven studies.

Table 2 summarizes types of outcome measurement techniques used and outcomes for TST, SL and SE in the included studies. Fifteen studies described subjective outcomes, among which five studies had both subjective and objective measures. Four studies reported descriptive sleep quality only, and are therefore excluded from the meta-analysis [25,29,30,34]. 

### 3.2. Total Sleep Time, TST

The forest plots as shown in Figure 2 (objective measures) examine the effect of melatonin and/or ramelteon on TST. For objectively measured TSTs (*n* = 607), the overall SMD was moderate (0.16, *p* = 0.04), but variability between studies as indicated by I^2^ was low (0%); hence, a fixed effect model was used. Jun et al., a RCT with prolonged release melatonin 2.0 mg had the largest SMD of 0.46 but did not show statistical significance (*p* = 0.37) [29]. Another study Jha et al. that used ramelteon 8.0 mg among insomniacs with heartburn symptoms over 4 weeks had the smallest mean difference (SMD = −0.42; *p* = 0.41) [28]. Publication bias was a problem, as seen from the funnel plot and the Kendall’s tau was not significant (*p* = 0.92).

In the subjectively measured TST forest plot (Appendix A, seven studies were included with a total sample size of 1545. The overall SMD (0.15, *p* = 0.38) was small and not significantly different than null. The variability between studies as indicated by I2 was high (91.3%), and one research group, Roth et al. [37,38] provided most of the data. The largest SMD was 1.63 with statistical significance and this 2006 study was an open-label study with menopausal women using ramelteon 8.0 mg [24]. A randomized crossover trial by Baskett et al. had the smallest mean difference of SMD −4.82 and it was statistically significant, favoring the placebo over the intervention. Publication bias was not a problem from the funnel plot while significant Kendall’s tau (*p* < 0.001) was seen [23].

### 3.3. Sleep Latency, SL

The forest plot for the eight studies examining the effect of melatonin and/or ramelteon on objective SL is shown in Figure 2. The total sample size was 1110; the overall SMD from the random effect model was 0.74 and was statistically significant (*p* < 0.001). There was a statistically significant mean difference for objectively measured SL among three studies: Haimov et al. [28], Penn Takeda et al. [34] and Roth et al. [38]. There was a large variability (I^2^ = 94.5%) between the studies. There was no evidence of publication bias in the funnel plot, and Kendall’s tau was significant (*p* < 0.001).

Eight studies reported subjective SL (Appendix A), and overall sample size was 3135, with a random effect model showing moderate SMD (0.40, *p* < 0.001). The mean difference was not statistically significant for two of the studies: Wade et al. [40], Roth et al. [38], The largest SMD was seen in Baskett et al. [25], which was statistically significant, while the study by Wade et al. [40] had the smallest SMD of 0.07 and *p* = 0.50. There was a high variability (I^2^ = 86.6%) and no evidence of publication bias in the funnel plot, with a significant Kendall’s tau (*p* < 0.001).

### 3.4. Sleep Efficiency, SE

The forest plot for objective SE is shown in Figure 2. There are eight studies with a total sample size of 901, and a moderate SMD of 0.41, (*p* = 0.07), but the variability as identified by I^2^ was very high (89.1%), indicating that there were functional differences between studies. Examination of the mean differences showed that there are two studies (Almeda et al. [23]; Baskett et al. [25]) that have negative SMD favoring placebo over melatonin and/or ramelteon. Only two other studies (Haimov et al. [28], and Zhadanova et al. [42]) showed statistically significant *p*-values and a large positive SMD. Since both are older studies, the quality of reporting could have influenced the large SMDs seen and further pushed the overall SMD of the forest plot favoring treatment with large positive value. There was a large variability (I^2^ = 89.1%) between the studies. There was no evidence of publication bias in the funnel plot; Kendall’s tau was significant (*p* < 0.001). Only one study, Jha et al., reported subjective SE and no forest plot was constructed [29].

### 3.5. Sleep Quality

Ten studies were included in the systematic review looking at sleep quality outcomes using melatonin and/or ramelteon over placebo (Appendix A). Sleep quality was measured using a variety of tools such as: sleep diaries/self report questionnaires [13,26,31,38,40,41], sleep logs with visual analogue scale [23,27], Pittsburgh Sleep Quality Index (PSQI) [25,36,40,41], or Leeds Sleep Evaluation Questionnaire (LSEQ) [40] in these studies. Four studies reported subjective sleep quality using questionnaires such as PSQI and/or LSEQ [25,36,40,41] and four reported using sleep diaries [13,31,40,41], while four others used self-report questionnaire [23,26,27,38] with varying numbers of items and scales. Through these different measurement methods, melatonin and/or ramelteon showed significant improvements on patient reported sleep quality. A single study showed a worse PSQI score in the intervention group than the placebo group, indicating melatonin did not significantly improve sleep quality [36].

### 3.6. Bias Assessment

Risk of bias findings of the 16 studies included in the meta-analysis with RCT study design are presented in Table 3. Overall, there were five studies with high risk of bias, four with some concerns, and seven with low risk of bias. Randomization bias factor was the domain showing the lowest risk. The missing outcome data domain showed the highest risk with three studies showing some concerns and four studies having high risk of bias. 

## 4. Discussion

This systematic review with meta-analysis represents the latest and most comprehensive search and synthesis of literature on the effects of melatonin or the melatonin receptor agonist, ramelteon, on sleep outcomes in older patients with chronic insomnia. To our knowledge, this is the most up-to-date meta-analysis to provide evidence of efficacy of melatonin and/or ramelteon in TST, SL and SE measured objectively and subjectively among older adults in various settings. The most important finding of this study is that melatonin and/or ramelteon produced significantly higher benefit over placebo for increasing TST, measured objectively, with an average of 21 min longer. Based on objective measurements, subjects assigned to melatonin and/or ramelteon fell asleep an average of 13.8 min earlier than those receiving placebo. While in subjective measurements, sleep latency was reduced significantly with 8.3 min in then melatonin and/or ramelteon group compared to the placebo. Sleep efficiency was not statistically significantly improved by melatonin and/or ramelteon use. The effect (overall SMDs) showed inconsistency within the included studies resulting in functionally non-homogeneous groups. Potential reasons for this variation may include varied doses of melatonin and/or ramelteon used, fluctuating duration of treatments, and different sample sizes.

Although several systematic reviews and meta-analyses of melatonin and/or ramelteon testing their effects on sleep outcomes were performed previously, none were conducted recently, specifically assessed the impact on older adults, evaluated both melatonin and ramelteon, or included comparative studies outside of RCTs. For example, Brzezinski et al. [43] conducted a meta-analysis focused on objective sleep outcomes reported in RCTs only with search results ending in 2003. Similarly, meta-analysis by Buscemi et al. [11] included studies from 1999–2003 that targeted efficacy of melatonin published in RCTs only. This study was also not specific to older adults and was looking at secondary sleep disorders. Liu et al. [44] is a systematic review and a meta-analysis that included studies published up to 2011 and only included RCTs focused on ramelteon only versus placebo among adults. While a 2014 meta-analysis [17] addressed efficacy and safety of ramelteon in adults and another Ferracioli-Oda et al. [19] is a meta-analysis that addressed melatonin usage in adults and children with primary sleep disorders, neither search went beyond 2012. In comparison, the current systematic review and meta-analyses included all studies using comparative design involving older adults, a high-risk population with insomnia with searches conducted to 2021.

As expected, there was substantial variability between the included studies concerning study design, sample size, intervention (doses and duration), and methods to measure outcomes. Studies with small sample size may show larger treatment effects on certain sleep outcomes, and selection bias may have played a role in some of the sleep outcomes such as publication bias seen in TST. Although all included subjects in the current study were older (mean age 55+ years), the age range varied among the included studies from 58.8 to 82.9 years. The sleep architecture might vary among different age ranges such as 55–75 vs. those above 75 years old. Study settings also varied, ranging from outpatient to long term care, which may mean health status of included subjects may have varied. Similarly, there were wide differences in the duration over which outcomes were measured and doses of melatonin and/or ramelteon used in the studies. The duration of therapy ranged from one week to 52.1 weeks. With no established standard amount, melatonin dose varied from 0.3 mg to 5.4 mg, while ramelteon had fewer dose options with either 4 mg or 8 mg used. Thus, the forest plots should be interpreted with caution as longer treatment durations can impact significance of results.

There were differences in how the outcomes were measured with majority of studies reporting objective sleep outcomes. Objective TST were found to be significantly improved in our study, while previous studies showed mixed results, with three meta-analyses [19,42,43] reporting increased TST and one meta-analysis reported no associated improvement in TST [16]. Consistent with previous systematic reviews and meta-analyses [43,44,45], our study found significant reductions in SL measured both objectively and subjectively. One unique observation seen in our analysis was that the subjective TST and SL always reported lower SMD compared with objective outcomes, suggesting that perceived sleep outcomes are worse than objective measures. Interestingly, a study aimed to look at this disagreement between subjective and actigraphy measures of sleep duration among older adults with mean age of 68 years [46]. This study showed that poor sleepers tended to consistently report shorter TST in their diaries than was measured by actigraphy. The difference in outcome was dependent on age, gender, subjective sleep quality, the use of sleep medication, depressive symptoms, poor cognitive function, and functional disability [46].

There were potential limitations to the present study. As noted above, some of the included studies were of low methodological quality; thus, findings need to be interpreted with caution. Given that insomnia is a chronic disorder, the study durations of included trials were relatively short, with a mean length of 46 days. Moreover, most of the included RCTs used a crossover design, and a period effect in these studies cannot be excluded, although this could be a minimal bias/muted effect. Another potential limitation is that reported doses of melatonin products may not accurately reflect their contents, since the strength and purity of melatonin is not regulated by the United States Food and Drug Administration, so there is a possibility that some of the products studied may deviate from what is on the labeling [47]. Due to a limited number of ramelteon studies among older adults, we were unable to conduct subgroup analyses comparing melatonin versus ramelteon. Likewise, subgroup analyses per study design was not performed due to limited studies using a non-RCT design. Despite these limitations, our study demonstrated that melatonin and/or ramelteon administered to older subjects with chronic insomnia modestly improved objective TST, SL and subjective sleep quality. Sleep efficiency was not significantly different.

## 5. Conclusions

With limited safe treatment options for insomnia in older adults, the current study supports evidence about the moderate efficacy of melatonin and a melatonin receptor agonist (ramelteon) in increasing total sleep time and reducing sleep latency. With widely known concerns about the safety of sleep agents, such as benzodiazepines and nonbenzodiazepine benzodiazepine-receptor agonists, melatonin and/or ramelteon may be safe and effective options for older patients with insomnia.

## Figures and Tables

**Figure 1 jcm-11-05138-f001:**
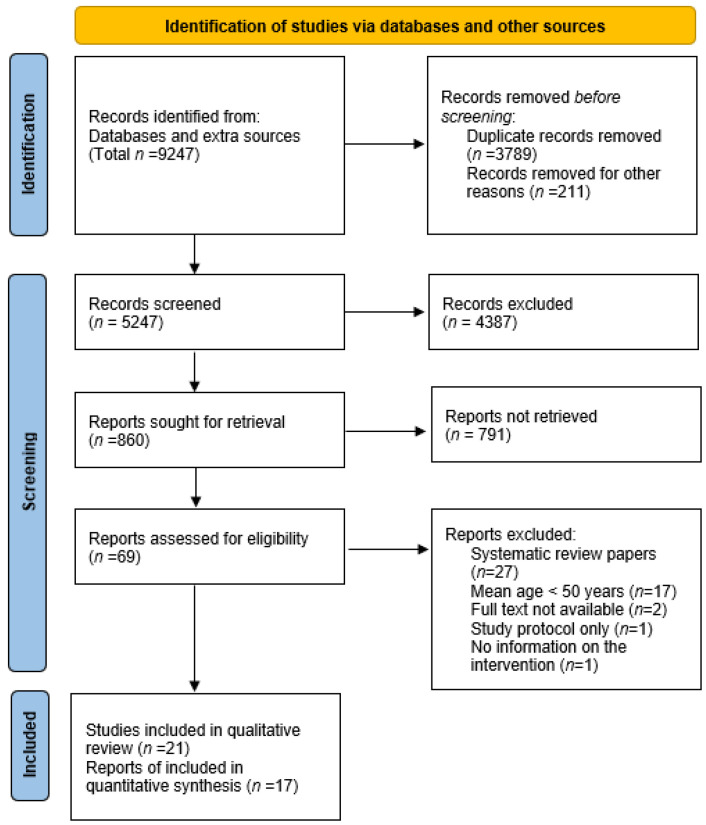
PRISMA flowchart.

**Figure 2 jcm-11-05138-f002:**
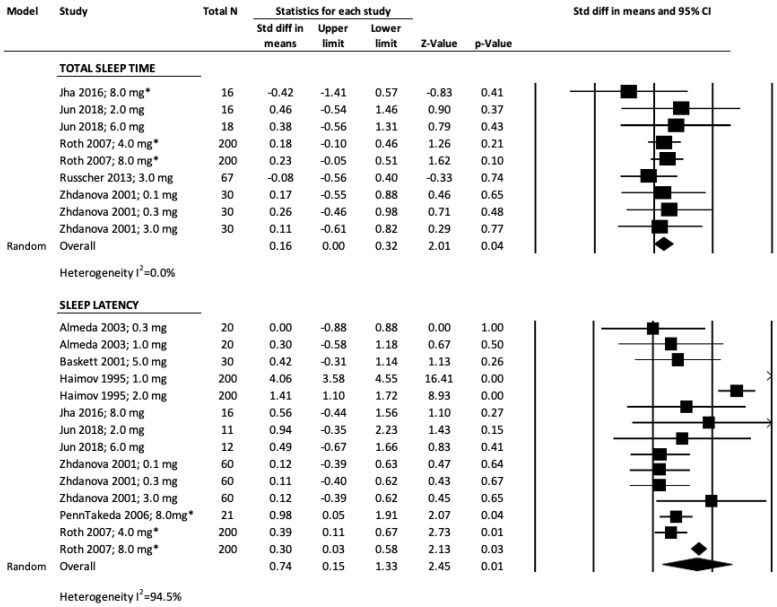
Objective Sleep Outcomes. The columns of the figures as labeled above in the figure represents: model, study, total N, standardized difference in means, upper limit, lower limit, Z-value, *p*-value, standardized difference in means and 95% CI.

**Table 1 jcm-11-05138-t001:** Characteristics of the Included Studies in the systematic review.

Study Author, Year	Country	Study Design	Total N	Patient Age, Years Mean ± SD	Male, %	Female, %	Study Settings	Concurrent Disease	Duration of Therapy, Days	Drug, Dose (mg)
Andrade et al., 1999 [24]	India	RCT	33	55.6 ± 12.7	73	27	Inpatient	None	8–16	Melatonin, 5.4
Haimov et al., 1995 [28]	Israel	RCT	51	75.2 ± 6	57	43	Inpatient & Outpatient	None	70	Melatonin 1.0, 2.0
Jha et al., 2016 [29]	USA	RCT	16	53 ± 4.27	88	12	Inpatient	Gastroesophageal Reflux	28	Ramelteon, 8.0
Jun et al., 2018 [30]	Korea	RCT	25	66.4 ± 8.64	64	36	Inpatient	iRBD	28	Melatonin, 2.0, 6.0
Lemoine et al., 2007 [13]	France/Israel	RCT	170	68.5 ± 8.3	34	66	Outpatient	Cardiovascular conditions	21	Melatonin, 2.0
Mini, et al., 2007 [33]	USA	RCT	327	72.5 ± 5.98	39	61	Outpatient	None	35	Ramelteon, 8.0
Penn Takeda et al., 2006 [34]	USA	RCT	27	72 ± 5.6	70	30	Outpatient	None	28	Ramelteon, 8.0
Roth et al., 2006 [37]	USA	RCT	829	72.4 ± 5.95	41	59	Outpatient	None	35	Ramelteon, 4.0, 8.0
Roth et al., 2007 [38]	USA	RCT	100	70.7 (65–85)	37	63	Outpatient	None	63	Ramelteon, 4.0, 8.0
Russcher et al., 2013 [39]	The Netherlands	RCT	67	65.0 ± 11.9	62	38	Long term care	hemodialysis	365	Melatonin, 3.0
Wade et al., 2010 [41]	Europe	RCT	281	71.0 ± 4.1	35	65	Outpatient	None	21	Melatonin, 2.0
Wade et al., 2007 [40]	Europe	RCT	334	65.7 ± 6.4	40	60	Outpatient	None	21	Melatonin, 2.0
Zhdanova et al., 2001 [42]	USA	RCT	30	>50	N/A		Outpatient	Chronic insomnia	63	Melatonin, 0.1, 0.3, 3.0
Almeida et al., 2003 [23]	Mexico	Crossover	10	50 ± 12.7	60	40	Outpatient	None	7	Melatonin, 0.3, 1.0
Baskett et al., 2001 [25]	New Zealand	Crossover	34	71.7 ± 4.9	32	68	Healthy volunteers	None	84	Melatonin, 5
Neurim pharma, 1995 [32]	Israel	Crossover	36	63 ± 8	31	69	Outpatient	Diabetes Mellitus, Type 2	21	Melatonin, 2.0
Dobkin et al., 2006 [26]	USA	Open-label study	20	52 ± 4.89	0 (only women study)	100	Academic medical center	Menopausal women	42	Ramelteon, 8.0
Fainstein et al., 1997 [27]	Argentina	Open-label study	41	74 ± 1.2	32	68	Inpatient	Depression/Dementia	21	Melatonin, 3.0
Lemoine et al., 2011 [31]	France/Israel	Open-Label	96	55.3 ± 13.0	31	69	Outpatient	none	365	Melatonin, 2.0
Richardson et al., 2009 [35]	USA	Open-label study	248	72.3 ± 5.6	47 ^a^	53	Outpatient	None	336	Ramelteon, 8.0
Rondanelli et al., 2011 [36]	Italy	Open-label study	43	78.3 ± 3.9	37	63	Long term care	None	60	Melatonin, 5.0

**Table 2 jcm-11-05138-t002:** Study Outcomes.

Study Author, Year	Outcome Measure	Total Sleep Time, min Mean ± SD		Sleep Latency, min Mean ± SD		Sleep Efficiency, % Mean ± SD	
		Treatment	Placebo	Treatment	Placebo	Treatment	Placebo
**Melatonin**							
Andrade et al., 1999 [24]	15-item structure sleep questionnaire	354 ± 54	300 ± 96	18 ± 12	60 ± 60		
Haimov et al., 1995 [28]	Actigraphy			1.0 mg: 14 ± 5.0 2.0 mg: 37 ± 11.0	54 ± 13.0	1.0 mg: 84.3 ± 2.3 2.0 mg: 80.41 ± 1.8	77.4 ± 1.9
Jun et al., 2018 [30]	PSG, self-questionnaire	2.0 mg: 399.4 ± 58.5 6.0 mg: 398.3 ± 73.9	374.5 ± 50.4	2.0 mg: 20.7 ± 9.5 6.0 mg: 25.9 ± 40.2	13.1 ± 7.3	2.0 mg: 79.7 ± 10 6.0 mg: 79.6 ± 12.9	72.8 ± 7.1
Russcher et al., 2013 [39]	Actigraphy, QoL questionnaire, Melatonin in saliva, Ambulatory blood pressure, echocardiography	318 ± 29	323 ± 82	Median (IQR) 20.3 (30)	Median (IQR) 25.0 (31)	66.3 ± 19.7	64.9 ± 18.1
Wade et al., 2010 [41]	Sleep diary, PSQI	Change in subjective total sleep time from baseline: 20.4 ± 45	Change in subjective total sleep time from baseline: 12 ± 47.4	Change in subjective sleep latency from baseline: −19.1 ± 47.3	Change in subjective sleep latency from baseline: −1.7 ± 47.8		
Wade et al., 2007 [40]	PSQI, LSEQ, sleep diary			Subjective sleep latency 40.8 ± 54.5	Subjective sleep latency 45 ± 59		
Zhdanova et al., 2001 [42]	Polysomnography, wrist reports, Actigraphy, Electrocardiography	0.1 mg: 402 ± 45 0.3 mg: 409 ± 49 3.0 mg: 398 ± 56	390 ± 91	0.1 mg: 10 ± 6 0.3 mg: 10 ± 8 3.0 mg: 10 ± 7	11 ± 10	0.1 mg: 84 ± 8 0.3 mg: 88 ± 7 3.0 mg: 84 ± 8	78 ± 15
Almeida et al., 2003 [23]	EEG, Sleep logs with analogue visual scale	0.3 mg: 380 1.0 mg: 375	400	0.3 mg: 69.2 ± 29.1 1.0 mg: 57.4 ± 47.2	69.2 ± 29.1	0.3 mg: 84 1.0 mg: 82	84
Baskett et al., 2001 [25]	PSQI, Actigraphy	438 (432, 456) PSQI subjective measures: 340 (290-400)	444 (420,468) 443 (410,60)	1.6 (0.6, 2.8) 10 (10-20)	1.4 (0.4, 2.0) 10 (5-15)	84.1 (83.8, 86.3) 69 (55-77)	86.2 (84.9, 87.1) g 91 (86-93)
Neurim pharma, 1995 [32]	Wrist actigraphy					83.1 ± 11.3	79.5 ± 9.6
**Ramelteon**							
Jha et al., 2016 [29]	PSQI, sleep diaries, actigraphy	430.95 ± 95.67	466.07 ± 69.49	9.64 ± 38.71	27.59 ± 23.28	87	83
Mini, et al., 2007 [33]	Sleep diaries			Change in subjective sleep latency from baseline −37.4	Change in subjective sleep latency from baseline −17.1		
Penn Takeda et al., 2006 [34]	Polysomnography			9.7 ± 10.3	34.4 ± 30.7		
Roth et al., 2006 [37]	Sleep diaries	4.0 mg: 337.5 8.0 mg: 334.4	4.0 mg: 330.1 8.0 mg: 330.1	4.0 mg: 63.4 8.0 mg: 57.7	4.0 mg: 70.6 8.0 mg: 70.6		
Roth et al., 2007 [38]	Polysomnography; Post Sleep Questionnaire	4.0 mg: 359.4 (50.6) 8.0 mg: 362.0 (50.3) Subjective scores; 4.0 mg: 337.8 (66.8) 8.0 mg: 337.0 (66.2)	350.4 (50.4) Subjective score; 333.9 (66.9)	4.0 mg: 28.7 (24.9) 8.0 mg: 30.8 (25.2) Subjective scores; 4.0 mg: 48.2 (45.3) 8.0 mg: 50.9 (44.6)	38.4 (24.9) Subjective score; 58.2 (45.3)	4.0 mg: 74.9 (10.5) 8.0 mg: 75.5 (10.5)	73.1 (10.5)
Dobkin et al., 2006 [26]	Sleep diaries and self-report questionnaires *	420 ± 38	336 ± 62	24.0 ± 15.0	46.2 ± 19.8	91 ± 6	80 ± 10
Richardson et al., 2009 [35]	Sleep diaries	370	350	42	50		

* Change from baseline reported.

**Table 3 jcm-11-05138-t003:** Risk of bias.

Study Author, Year	Randomization	Deviations from the Intended Intervention	Missing Outcome Data	Measurement of Outcome	Selection of the Reported Results	Overall
Almeida et al., 2003 [23]						
Andrade et al., 1999 [24]				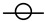		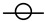
Baskett et al., 2001 [25]						
Haimov et al., 1995 [28]		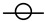	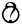			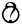
Jha et al., 2016 [29]						
Jun et al., 2018 [30]						
Neurim pharma, 1995 [32]			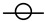			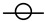
Mini, et al., 2007 [33]			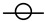	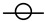	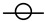	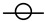
Pen state Takeda et al., 2006 [34]			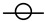			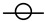
Rondanelli et al., 2011 [36]						
Roth et al., 2006 [37]			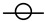			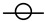
Roth et al., 2007 [38]						
Russcher et al., 2013 [39]						
Wade et al., 2010 [41]	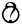	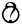				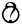
Wade et al., 2007 [40]			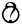			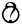
Zhdanova et al., 2001 [42]			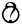			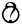

High risk of bias = 
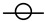
, Low risk of bias = 

, Some concern = 
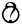
.

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
