# Peer review of "Use of Melatonin and/on Ramelteon for the Treatment of Insomnia in Older Adults: A Systematic Review and Meta-Analysis"

_jcm, 2022, doi:10.3390/jcm11175138_

Round 1
Reviewer 1 Report
The manuscript is interesting and as you know based on this meta-analysis result the policymakers and other studies may take clinical decisions on the "Use of Melatonin and Ramelteon for the Treatment of Insomnia in Older Adults", therefore, there are some points that should be taken to your concern which affects the results of the systematic review and meta-analysis:
1- Please write a separate section for the inclusion and exclusion criteria following PICOs methods or any other method for more clarity.
2- In the methodology section you mention that" RCT, prospective or retrospective cohort, case-control, or other observational study design with com- 100 parative groups that tested the effects of melatonin and/or ramelteon on insomnia 101 outcomes in older adults" which means there is heterogeneity in the included studies which have a different design, different characteristics, outcomes, measurements, and sure it has also different tool/method for the bias and quality assessment while here you have used the same tool for evaluating the whole included studies which is not correct from my view and it's affecting your results. Moreover, you should do the bias and quality assessment before the meta-analysis, and only studies with a low risk of bias to be included in the meta-analysis. Therefore, you should address all these comments, or it's enough to do a systematic review with qualitative synthesis without meta-analysis.
3- You mention that you have included only the English studies and this is a selection bias! , also in the study characteristic, you have included only the male! how about the females?
4- In the included study please include the study outcomes and measurements.
5- In the PRISMA flowchart you mention that you excluded studies for reasons please explain in detail for each number in each step what is the reasons, and please follow the PRISMA checklist to report your systematic review.
6- As one of the characteristics of the systematic review and meta-analysis has a reproducible methodology, therefore here are some of the published systematic reviews and meta-analyses which may help you in your revision of how to make it more clear and consistent, and on the right path for this research which consider one of the top research and the results should be with highest reliability and validity:
- https://doi.org/10.3390/ijerph18157765
- https://doi.org/10.1111/raq.12605
7- You have mentioned that "A comprehensive review of the literature, supplemented by additional grey literature and relevant sleep journal article searches, was conducted by the primary author", It's not methodological correct to be reviewed from only one author and it will have biases, and how about if there is a conflict in the decision of the studies to be included or excluded how you will treat that please revise and explain in detail.
Author Response
Comments and Suggestions for Authors
-Reviewer 1
INTRODUCTION It would be worth pointing out that melatonin come in two forms. Simple melatonin has a short half life and tend to be used to treat circadian rhythm disorders. Slow release melatonin has a longer half life and is used to treat insomnia. Sleep Res 2007; 16:372. BMC Med 2010; 8(5). Curr Res Med Opin 2010; 27(1): 87. ​
Thank you for the comments and authors have included a sentence to mention about the two types of available formulations. Its included as a sentence in introduction in the line 67 and one of the article that states this information is already included in our SR with reference number 32 and thus included after this new sentence.
The use of the terms melatonin/ramelteon and melatonin and/or ramelteon is confusing, as it implies that both drugs were used together, but this does not seem to be the case when looking at Table 1. please clarify.
Thank you for your comments and the concern is genuine. Our intention was to tell the reader that in some cases its can be both melatonin and ramelteon and in some other cases it could be either melatonin or ramelteon. So would use consistent term “melatonin and/or ramelteon” and we changed it throughout the manuscript and tables to reflect this.
It is not clear why have "older adults" in the title and other parts of the manuscript when the mean age of subjects was 55.
Thanks for your comment and our main focus of this study is to understand the use of melatonin and/or ramelteon in geriatric population as they are the one most affected by insomnia. Like we defined geriatric population as those aged 65 years and older, we included studies which reported mean age of 50 years or older to broaden the search results of the systematic review as suggested by the librarian expert in conducting and designing such reviews. This explanation and reason for including mean age of 50 years is clearly stated in the methods in line 102-103.
DISCUSSION It would be useful to summarize the results for readers explaining effect sizes in minutes especially for TST and sleep latency.
Totally agree with your suggestion and accordingly we used the effect sizes summarized in minutes for explaining TST and sleep latency as they only these both outcomes were statistically significant and included the change or improvement in the discussion section first paragraph In line 375.
MINOR POINTS
Lines 40 -42 - the wording of this sentence is awkward making it difficult to read.
We agree with this comment and accordingly changes were made to this sentence to improve it comprehensiveness.
Comments and Suggestions for Authors
- Reviewer 2
The manuscript is interesting and as you know based on this meta-analysis result the policymakers and other studies may take clinical decisions on the "Use of Melatonin and Ramelteon for the Treatment of Insomnia in Older Adults", therefore, there are some points that should be taken to your concern which affects the results of the systematic review and meta-analysis:
- Please write a separate section for the inclusion and exclusion criteria following PICOs methods or any other method for more clarity.
The inclusion and exclusion criteria followed in this systematic literature review was clearly given under section “2.1 study selection” whereby all criteria used for including studies as well as exclusion were reported. The authors confirm that we were well aware and used the PICOS framework to write this section. For more clarity, changes were made such that it is evidently mentioned in the manuscript.
- In the methodology section you mention that" RCT, prospective or retrospective cohort, case-control, or other observational study design with com- 100 parative groups that tested the effects of melatonin and/or ramelteon on insomnia 101 outcomes in older adults" which means there is heterogeneity in the included studies which have a different design, different characteristics, outcomes, measurements, and sure it has also different tool/method for the bias and quality assessment while here you have used the same tool for evaluating the whole included studies which is not correct from my view and it's affecting your results. Moreover, you should do the bias and quality assessment before the meta-analysis, and only studies with a low risk of bias to be included in the meta-analysis. Therefore, you should address all these comments, or it's enough to do a systematic review with qualitative synthesis without meta-analysis.
Yes, we totally agree with reviewers’ comments that there are different study designs. Characteristics of the studies and outcomes measured with varying methods. Keeping that in mind we have well conducted the risk of bias assessment for this systematic review, and it was conducted before running the meta-analysis. It is clearly explained in the methods section in line 173 about the tool used to measure bias. Changes were made to the sentence to better reflect this.
All studies captured in the meta-analysis were of RCT study design with either cross-over or open-study label and they have been appropriately assessed using the following domains of the Cochrane Collaboration’s tool were selected to evaluate the risk of bias: arising from the randomization process; from assignment to intervention; from missing outcome data; measurement of the outcome; and selection of the re-ported results. The intention of conducting risk of bias assessment of these studies were to assess the quality of studies and if possible, also do a sub group analysis in the quantitative study of meta-analysis to see how low and high risk of bias can impact the overall effect size and direction of the standardized difference in means. However, based on the low and high risk of bias, there were not sufficient number of studies for each outcome assessed in meta-analysis to conduct a subgroup analysis thus this was further not conducted. Overall only a summary of risk of bias of the studies were included for readers knowledge and judgment. All studies included in both systematic review and meta-analysis underwent risk of bias assessment.
We believe this meta-analysis provides lot of useful information about effect size and direction of the effects associated with melatonin and/or ramelteon and thus strongly believe this must be included along with qualitative analysis.
3- You mention that you have included only the English studies and this is a selection bias! , also in the study characteristic, you have included only the male! how about the females?
Unfortunately, we could not include any studies published in other languages than English due to the inability to read and understand any other languages by the authors. Thus, only studies published in English language was included in our systematic review. In the study characteristics just to adjust for number of columns we only included males. On the basis of the reviewer comment, we have included the females as an additional column in table 1. Thanks
4- In the included study please include the study outcomes and measurements.
Thanks for your comments, in all the 17 included studies of the meta-analysis we have the outcomes measured with values along with methods use to measure the outcomes in table 2-clearly tilted as “study outcomes”. The authors believe the table is comprehensive enough to give readers information about the study outcomes and measurements. Also detailed information about study outcomes and measurements are provided in methods under “outcomes” section in line 150.
5- In the PRISMA flowchart you mention that you excluded studies for reasons please explain in detail for each number in each step what is the reasons, and please follow the PRISMA checklist to report your systematic review.
Thank you for this. As per your suggestion we used the latest PRISMA flowchart and guidelines. As per the PRISMA checklist we have included all the reasons of exclusions and the number of articles excluded with reasons at each stage of the review. The figure 1 of our manuscript was modified to this change.
6- As one of the characteristics of the systematic review and meta-analysis has a reproducible methodology, therefore here are some of the published systematic reviews and meta-analyses which may help you in your revision of how to make it more clear and consistent, and on the right path for this research which consider one of the top research and the results should be with highest reliability and validity:
- https://doi.org/10.3390/nu12020521
- https://doi.org/10.3390/nu13031028
- https://doi.org/10.3390/ijerph18157765
- https://doi.org/10.1111/raq.12605
- https://doi.org/10.3390/ijerph18010263.
Thank you to the reviewer 2 for sharing these links. We have gone through all these articles methodology and reworked on our methods section to make it clearer and more consistent.
7- You have mentioned that "A comprehensive review of the literature, supplemented by additional grey literature and relevant sleep journal article searches, was conducted by the primary author", It's not methodological correct to be reviewed from only one author and it will have biases, and how about if there is a conflict in the decision of the studies to be included or excluded how you will treat that please revise and explain in detail.
Thank you for your comment, we agree with your statement and in reality two independent reviewers have assessed all study titles, abstracts, and full-text articles independently and met to combine the data by consensus. A third reviewer with clinical expertise in geriatric practice served as an arbiter when consensus was unable to be reached. This was also clearly reported under “data extraction” in line 141. Accordingly, the results section was changed to reflect how review of literature was conducted by two reviewers independently.
Reviewer 2 Report
INTRODUCTION It would be worth pointing out that melatonin come in two forms. Simple melatonin has a short half life and tend to be used to treat circadian rhythm disorders. Slow release melatonin has a longer half life and is used to treat insomnia. Sleep Res 2007; 16:372. BMC Med 2010; 8(5). Curr Res Med Opin 2010; 27(1): 87. ​
The use of the terms melatonin/ramelteon and melatonin and/or ramelteon is confusing, as it implies that both drugs were used together, but this does not seem to be the case when looking at Table 1. please clarify.
It is not clear why have "older adults" in the title and other parts of the manuscript when the mean age of subjects was 55.
DISCUSSION It would be useful to summarize the results for readers explaining effect sizes in minutes especially for TST and sleep latency.
MINOR POINTS
Lines 40 -42 - the wording of this sentence is awkward making it difficult to read.
Author Response
Comments and Suggestions for Authors
- Reviewer 2
The manuscript is interesting and as you know based on this meta-analysis result the policymakers and other studies may take clinical decisions on the "Use of Melatonin and Ramelteon for the Treatment of Insomnia in Older Adults", therefore, there are some points that should be taken to your concern which affects the results of the systematic review and meta-analysis:
- Please write a separate section for the inclusion and exclusion criteria following PICOs methods or any other method for more clarity.
The inclusion and exclusion criteria followed in this systematic literature review was clearly given under section “2.1 study selection” whereby all criteria used for including studies as well as exclusion were reported. The authors confirm that we were well aware and used the PICOS framework to write this section. For more clarity, changes were made such that it is evidently mentioned in the manuscript.
- In the methodology section you mention that" RCT, prospective or retrospective cohort, case-control, or other observational study design with com- 100 parative groups that tested the effects of melatonin and/or ramelteon on insomnia 101 outcomes in older adults" which means there is heterogeneity in the included studies which have a different design, different characteristics, outcomes, measurements, and sure it has also different tool/method for the bias and quality assessment while here you have used the same tool for evaluating the whole included studies which is not correct from my view and it's affecting your results. Moreover, you should do the bias and quality assessment before the meta-analysis, and only studies with a low risk of bias to be included in the meta-analysis. Therefore, you should address all these comments, or it's enough to do a systematic review with qualitative synthesis without meta-analysis.
Yes, we totally agree with reviewers’ comments that there are different study designs. Characteristics of the studies and outcomes measured with varying methods. Keeping that in mind we have well conducted the risk of bias assessment for this systematic review, and it was conducted before running the meta-analysis. It is clearly explained in the methods section in line 173 about the tool used to measure bias. Changes were made to the sentence to better reflect this.
All studies captured in the meta-analysis were of RCT study design with either cross-over or open-study label and they have been appropriately assessed using the following domains of the Cochrane Collaboration’s tool were selected to evaluate the risk of bias: arising from the randomization process; from assignment to intervention; from missing outcome data; measurement of the outcome; and selection of the re-ported results. The intention of conducting risk of bias assessment of these studies were to assess the quality of studies and if possible, also do a sub group analysis in the quantitative study of meta-analysis to see how low and high risk of bias can impact the overall effect size and direction of the standardized difference in means. However, based on the low and high risk of bias, there were not sufficient number of studies for each outcome assessed in meta-analysis to conduct a subgroup analysis thus this was further not conducted. Overall only a summary of risk of bias of the studies were included for readers knowledge and judgment. All studies included in both systematic review and meta-analysis underwent risk of bias assessment.
We believe this meta-analysis provides lot of useful information about effect size and direction of the effects associated with melatonin and/or ramelteon and thus strongly believe this must be included along with qualitative analysis.
3- You mention that you have included only the English studies and this is a selection bias! , also in the study characteristic, you have included only the male! how about the females?
Unfortunately, we could not include any studies published in other languages than English due to the inability to read and understand any other languages by the authors. Thus, only studies published in English language was included in our systematic review. In the study characteristics just to adjust for number of columns we only included males. On the basis of the reviewer comment, we have included the females as an additional column in table 1. Thanks
4- In the included study please include the study outcomes and measurements.
Thanks for your comments, in all the 17 included studies of the meta-analysis we have the outcomes measured with values along with methods use to measure the outcomes in table 2-clearly tilted as “study outcomes”. The authors believe the table is comprehensive enough to give readers information about the study outcomes and measurements. Also detailed information about study outcomes and measurements are provided in methods under “outcomes” section in line 150.
5- In the PRISMA flowchart you mention that you excluded studies for reasons please explain in detail for each number in each step what is the reasons, and please follow the PRISMA checklist to report your systematic review.
Thank you for this. As per your suggestion we used the latest PRISMA flowchart and guidelines. As per the PRISMA checklist we have included all the reasons of exclusions and the number of articles excluded with reasons at each stage of the review. The figure 1 of our manuscript was modified to this change.
6- As one of the characteristics of the systematic review and meta-analysis has a reproducible methodology, therefore here are some of the published systematic reviews and meta-analyses which may help you in your revision of how to make it more clear and consistent, and on the right path for this research which consider one of the top research and the results should be with highest reliability and validity:
- https://doi.org/10.3390/nu12020521
- https://doi.org/10.3390/nu13031028
- https://doi.org/10.3390/ijerph18157765
- https://doi.org/10.1111/raq.12605
- https://doi.org/10.3390/ijerph18010263.
Thank you to the reviewer 2 for sharing these links. We have gone through all these articles methodology and reworked on our methods section to make it clearer and more consistent.
7- You have mentioned that "A comprehensive review of the literature, supplemented by additional grey literature and relevant sleep journal article searches, was conducted by the primary author", It's not methodological correct to be reviewed from only one author and it will have biases, and how about if there is a conflict in the decision of the studies to be included or excluded how you will treat that please revise and explain in detail.
Thank you for your comment, we agree with your statement and in reality two independent reviewers have assessed all study titles, abstracts, and full-text articles independently and met to combine the data by consensus. A third reviewer with clinical expertise in geriatric practice served as an arbiter when consensus was unable to be reached. This was also clearly reported under “data extraction” in line 141. Accordingly, the results section was changed to reflect how review of literature was conducted by two reviewers independently.
Round 2
Reviewer 1 Report
It is very clear that you didn't address the comments that I have suggested and recommended for you which really very crucial in conducting any systematic review and it's very clear for anyone if he/she reads your manuscript and for future referral as well. I understand you tried to explain to me in your response to my comments in your response to the reviewer comments notes but your revision manuscript didn't improve according to the comments I raised to you.
Author Response
Comments and Suggestions for Authors
-Reviewer 1
INTRODUCTION It would be worth pointing out that melatonin come in two forms. Simple melatonin has a short half life and tend to be used to treat circadian rhythm disorders. Slow release melatonin has a longer half life and is used to treat insomnia. Sleep Res 2007; 16:372. BMC Med 2010; 8(5). Curr Res Med Opin 2010; 27(1): 87. ​
Thank you for the comments and authors have included a sentence to mention about the two types of available formulations. Its included as a sentence in introduction in the line 67 and one of the article that states this information is already included in our SR with reference number 32 and thus included after this new sentence.
The use of the terms melatonin/ramelteon and melatonin and/or ramelteon is confusing, as it implies that both drugs were used together, but this does not seem to be the case when looking at Table 1. please clarify.
Thank you for your comments and the concern is genuine. Our intention was to tell the reader that in some cases its can be both melatonin and ramelteon and in some other cases it could be either melatonin or ramelteon. So would use consistent term “melatonin and/or ramelteon” and we changed it throughout the manuscript and tables to reflect this.
It is not clear why have "older adults" in the title and other parts of the manuscript when the mean age of subjects was 55.
Thanks for your comment and our main focus of this study is to understand the use of melatonin and/or ramelteon in geriatric population as they are the one most affected by insomnia. Like we defined geriatric population as those aged 65 years and older, we included studies which reported mean age of 50 years or older to broaden the search results of the systematic review as suggested by the librarian expert in conducting and designing such reviews. This explanation and reason for including mean age of 50 years is clearly stated in the methods in line 108-109.
DISCUSSION It would be useful to summarize the results for readers explaining effect sizes in minutes especially for TST and sleep latency.
Totally agree with your suggestion and accordingly we used the effect sizes summarized in minutes for explaining TST and sleep latency as they only these both outcomes were statistically significant and included the change or improvement in the discussion section first paragraph In line 376.
MINOR POINTS
Lines 40 -42 - the wording of this sentence is awkward making it difficult to read.
We agree with this comment and accordingly changes were made to this sentence to improve it comprehensiveness.
Comments and Suggestions for Authors
- Reviewer 2
The manuscript is interesting and as you know based on this meta-analysis result the policymakers and other studies may take clinical decisions on the "Use of Melatonin and Ramelteon for the Treatment of Insomnia in Older Adults", therefore, there are some points that should be taken to your concern which affects the results of the systematic review and meta-analysis:
- Please write a separate section for the inclusion and exclusion criteria following PICOs methods or any other method for more clarity.
The inclusion and exclusion criteria followed in this systematic literature review was clearly given under section “2.1 study selection” whereby all criteria used for including studies as well as exclusion were reported. The authors confirm that we were well aware and used the PICOS framework to write this section. For more clarity, we included a sentence in the manuscript about the PICOS framework in line 103.
- In the methodology section you mention that" RCT, prospective or retrospective cohort, case-control, or other observational study design with com- 100 parative groups that tested the effects of melatonin and/or ramelteon on insomnia 101 outcomes in older adults" which means there is heterogeneity in the included studies which have a different design, different characteristics, outcomes, measurements, and sure it has also different tool/method for the bias and quality assessment while here you have used the same tool for evaluating the whole included studies which is not correct from my view and it's affecting your results. Moreover, you should do the bias and quality assessment before the meta-analysis, and only studies with a low risk of bias to be included in the meta-analysis. Therefore, you should address all these comments, or it's enough to do a systematic review with qualitative synthesis without meta-analysis.
Yes, we totally agree with reviewers’ comments that there are different study designs. Characteristics of the studies and outcomes measured with varying methods. Keeping that in mind we have well conducted the risk of bias assessment for this systematic review, and it was conducted before running the meta-analysis. It is clearly explained in the methods section in line 175 about the tool used to measure bias. Changes were made to the sentence to better reflect this.
All studies captured in the meta-analysis were of RCT study design with either cross-over or open-study label and they have been appropriately assessed using the following domains of the Cochrane Collaboration’s tool were selected to evaluate the risk of bias: arising from the randomization process; from assignment to intervention; from missing outcome data; measurement of the outcome; and selection of the re-ported results. The intention of conducting risk of bias assessment of these studies were to assess the quality of studies and if possible, also do a sub group analysis in the quantitative study of meta-analysis to see how low and high risk of bias can impact the overall effect size and direction of the standardized difference in means. However, based on the low and high risk of bias, there were not sufficient number of studies for each outcome assessed in meta-analysis to conduct a subgroup analysis thus this was further not conducted. Overall only a summary of risk of bias of the studies were included for readers knowledge and judgment. All studies included in both systematic review and meta-analysis underwent risk of bias assessment.
We believe this meta-analysis provides lot of useful information about effect size and direction of the effects associated with melatonin and/or ramelteon and thus strongly believe this must be included along with qualitative analysis.
3- You mention that you have included only the English studies and this is a selection bias! , also in the study characteristic, you have included only the male! how about the females?
Unfortunately, we could not include any studies published in other languages than English due to the inability to read and understand any other languages by the authors. Thus, only studies published in English language was included in our systematic review. In the study characteristics just to adjust for number of columns we only included males. On the basis of the reviewer comment, we have included the females as an additional column in table 1. Thanks
4- In the included study please include the study outcomes and measurements.
Thanks for your comments, in all the 17 included studies of the meta-analysis we have the outcomes measured with values along with methods use to measure the outcomes in table 2-clearly tilted as “study outcomes”. The authors believe the table is comprehensive enough to give readers information about the study outcomes and measurements. Also detailed information about study outcomes and measurements are provided in methods under “outcomes” section in line 152.
5- In the PRISMA flowchart you mention that you excluded studies for reasons please explain in detail for each number in each step what is the reasons, and please follow the PRISMA checklist to report your systematic review.
Thank you for this. As per your suggestion we used the latest PRISMA flowchart and guidelines. As per the PRISMA checklist we have included all the reasons of exclusions and the number of articles excluded with reasons at each stage of the review. The figure 1 of our manuscript was modified to reflect this change in page 6.
6- As one of the characteristics of the systematic review and meta-analysis has a reproducible methodology, therefore here are some of the published systematic reviews and meta-analyses which may help you in your revision of how to make it more clear and consistent, and on the right path for this research which consider one of the top research and the results should be with highest reliability and validity:
- https://doi.org/10.3390/nu12020521
- https://doi.org/10.3390/nu13031028
- https://doi.org/10.3390/ijerph18157765
- https://doi.org/10.1111/raq.12605
- https://doi.org/10.3390/ijerph18010263.
Thank you to the reviewer 2 for sharing these links. We have gone through all these articles methodology and reworked on our methods section to make it clearer and more consistent.
7- You have mentioned that "A comprehensive review of the literature, supplemented by additional grey literature and relevant sleep journal article searches, was conducted by the primary author", It's not methodological correct to be reviewed from only one author and it will have biases, and how about if there is a conflict in the decision of the studies to be included or excluded how you will treat that please revise and explain in detail.
Thank you for your comment, we agree with your statement and in reality two independent reviewers have assessed all study titles, abstracts, and full-text articles independently and met to combine the data by consensus. A third reviewer with clinical expertise in geriatric practice served as an arbiter when consensus was unable to be reached. This was also clearly reported under “data extraction” in line 141. Accordingly, the methods section was changed to reflect how review of literature was conducted by two reviewers independently in line 125.